# Exploring the barriers to, and importance of, participant diversity in early-phase clinical trials: an interview-based qualitative study of professionals and patient and public representatives

Robin Chatters [1], Munyaradzi Dimairo,[1] Cindy Cooper,[1] Shamila Ditta,[1] Jonathan Woodward,[1] Katie Biggs [1], Della Ogunleye,[2] Fiona Thistlethwaite,[3] Christina Yap [4], Alexander Rothman[5]

DO deceased

For numbered affiliations see end of article.

**Correspondence to**
Mr Robin Chatters;
r.chatters@sheffield.ac.uk

## ABSTRACT

**Objectives** To explore the importance of, and barriers to achieving, diversity in early-phase clinical trials.

**Design** Qualitative interviews analysed using thematic analysis.

**Setting and participants** Five professionals (clinical researchers and methodologists) and three patient and public representatives (those with experience of early-phase clinical trials and/or those from ethnic minority backgrounds) were interviewed between June and August 2022. Participants were identified via their institutional web page, existing contacts or social media (eg, X, formerly known as Twitter).

**Results** Professionals viewed that diversity is not currently considered in all early-phase clinical trials but felt that it should always be taken into account. Such trials are primarily undertaken at a small number of centres, thus limiting the populations they can access. Referrals from clinicians based in the community may increase diversity; however, those referred are often not from underserved groups. Referrals may be hindered by the extra resources required to approach and recruit underserved groups and participants often having to undertake 'self-driven' referrals. Patient and public representatives stated that diversity is important in research staff and that potential participants should be informed of the need for diversity. Those from underserved groups may require clarification regarding the potential harms of a treatment, even if these are unknown. Education may improve awareness and perception of early-phase clinical trials. We provide 14 recommendations to improve diversity in early-phase clinical trials.

**Conclusions** Diversity should be considered in all early-phase trials. Consideration is required regarding the extent of diversity and how it is addressed. The increased resources needed to recruit those from underserved groups may warrant funders to increase the funds to support the recruitment of such participants. The potential harms and societal benefits of the research should be presented to potential participants in a balanced but accurate way to increase transparency.

### STRENGTHS AND LIMITATIONS OF THIS STUDY

⇒ We qualitatively interviewed a diverse sample of five professionals (clinical researchers and methodologists) and three patient representatives to explore the importance of, and barriers to achieving, diversity in early-phase clinical trials.

⇒ Data saturation was not achieved due to a small sample size; rather, we looked to achieve 'information power', as conceived by Malterud et al, where the size of the study was determined by the amount of information the sample holds.

⇒ Interviewees were predominately based in the UK and had experience of cancer early-phase clinical trials—this therefore limited the conclusions that can be made regarding other disease areas, sectors and countries.

## INTRODUCTION

The importance of equality, diversity and inclusion (EDI) in clinical research is well documented, with the COVID-19 pandemic bringing into sharp focus the lack of inclusion of underserved groups in clinical trials.[1–5] The value of inclusion of such populations in late-phase trials (ie, in phases 3 and 4) is widely accepted, as treatments may affect diverse subsets of the population differently.[6] As such, there is guidance for researchers to consider groups routinely underserved in research when designing and conducting clinical trials.[7–9] There is evidence to suggest that diversity in such trials may be increased by improving staff competency through training and increased recruitment of staff from diverse backgrounds.[6]

There is a lack of guidance regarding the importance of diversity and how to incorporate underserved groups into early-phase clinical trials (EPCTs). Two reviews and a

Food and Drug Administration white paper provide guidance around increasing clinical trial diversity in the drug development process, but all lack guidance specific to EPCTs.[10–12] EPCTs are conducted in healthy volunteers or participants with a specific health condition and help us understand the feasibility of the therapeutic approach, the appropriate dose of the new treatment, how the drug should be administered, how the body responds to the treatment and side-effects. More so, they hugely inform the design and conduct of related later-phase trials. Due to their small size and often high-risk nature, EPCTs have specific challenges regarding the inclusion of diverse populations.

Despite the unknowns, the current lack of diversity in EPCTs is clear. EPCTs under-represent women,[13] older adults,[14–16] those with lower education levels,[17] those who live further away from hospitals,[17] those from deprived areas,[18] ethnic minorities[19 20] and, for those studies undertaken in majority English-speaking societies, individuals with limited English proficiency.[21] It has been shown that participants experiencing socioeconomic deprivation have worse outcomes in cancer trials than those not experiencing socioeconomic deprivation[22]; however, we do not know if this is the case for EPCTs due to the lack of data on diversity. The development and deployment of COVID-19 vaccines highlighted the role of medical mistrust in preventing individuals from participating in EPCTs, especially in those from ethnically diverse communities.[23 24]

There is a lack of interventions to address the inequalities within EPCTs.[20] Several studies have been undertaken to assess the barriers to participating in EPCTs, which include the information provided to potential participants,[25] expectations of a therapeutic benefit from the trial treatment,[26 27] lack of social and financial support,[28] and restrictive eligibility criteria.[29] However, these studies focus on general recruitment barriers, rather than those that prevent underserved populations from participating. Potential methods of increasing diversity include community outreach, increased diversity of staff, culturally tailored interventions, bilingual study teams[6 30 31] and helping elderly adults navigate barriers to enrolment.[32]

Previous research has not discussed the importance of the inclusion of underserved populations in EPCTs, or when it may be important. It may be the case, for certain conditions, treatments or contexts, that a focus on the representation of the population is less important and therefore diverse populations may not be required. For example, due to suspected homogeneity in how individuals may react to the treatment under investigation. However, excluding certain populations from such trials may lead to an inequitable health policy stacked against under-represented subpopulations. There is also evidence of a potential 'trial effect' of improved outcomes, greater adherence to clinical guidelines and evidence-based practice of clinicians and institutions that take part in trials.[33] However, the consequences for patient health are uncertain.

We therefore undertook a qualitative study of professionals with knowledge of EPCTs (ie, clinical researchers and methodologists) and patient and public representatives to explore in which situations diversity may be important in EPCTs, how diversity has been (and could be) incorporated into EPCTs and the barriers to inclusion in such trials.

## METHODS

### Scope and participant eligibility

We undertook semi-structured qualitative interviews with both professionals (ie, clinical researchers and methodologists with knowledge of EPCTs) and patient and public representatives (those with experience of EPCTs and/or those from ethnic minority backgrounds). Interviews with professionals explored the level of importance of diversity in EPCTs and when it may be appropriate, their experience of incorporating diversity into EPCTs and the challenges of doing so. Interviews with the patient and public representatives focused on the challenges of participating in EPCTs and methods to improve diversity in such trials.

### Recruitment of interview participants

Potential participants were identified by either searching institutional web pages or registries for individuals who are involved in EPCTs (eg, clinical trials units (CTUs) that are listed on the UK Clinical Research Collaboration CTU website as having expertise in phase 1 studies); asking study collaborators and other contacts for any individuals they are aware of who may be experienced in conducting EPCTs; and via social media (eg, X, formerly known as Twitter).

Potential participants were sent an email with a participant information sheet inviting them to participate in the study. To identify a purposive selection of participants for the interview, individuals were asked to complete a short survey to express their interest in the study which was administered using Qualtrics. The questionnaire requested key demographic data including location, age, ethnicity (patient and public representatives only), experience (professional or patient and public representative), clinical specialty (professionals only), area of expertise (professionals only) and gender. At the end of the survey, respondents were able to indicate whether they were interested in participating in an interview.

Participants were selected purposefully from those who completed the survey (n=22), maximising variation in key characteristics (area of expertise, ethnicity for the patient and public representatives only, years of experience and gender). Additional participants were approached via email without having to complete the survey first if they were thought to increase the diversity of the sample.

### Research ethics and participant consent process

Consent to participate in the semi-structured interviews was obtained via a consent form, which was completed by the participant prior to the interview, and after completing the survey (if applicable). The participant signed the consent

form using an electronic (typed or image of their signature inserted into the form) signature. The form was then countersigned by the researcher, and a copy of the completed consent form was emailed back to the participant.

## Sample size

Data saturation was not achieved due to limitations in funding. Rather, we looked to achieve 'information power', as conceived by Malterud et al,[34] where the size of the study was determined by the amount of information the sample holds, with studies encompassing narrow aims, dense specificity and strong dialogue having higher information power. We aimed to interview between 8 and 10 individuals.

## Study design and data collection

The study used a phenomenological framework, as interviewees had direct experience of the phenomenon under study. Two semi-structured topic guides (see online supplemental materials) were used (one for professionals and another for the patient and public representatives). These were developed by RC, with the aims of the study used as a starting point. A literature review was undertaken to inform the topic guide, identifying a lack of guidance specifically relating to EPCTs. As a result, literature regarding EDI in later-phase trials (III or IV) was used to inform the interview questions.[4 6 35] The topic guides were reviewed by members of the study team, including DO (a patient and public representative), but they were not piloted. The topic guide was iteratively modified after each interview. Seven of the eight interviews were conducted by RC (a male research fellow with a BSc), with assistance from JW (a male research assistant with an MSc) and SD (a female research assistant with an MA). One interview was conducted by SD and JW. All three interviewers had the experience of working on clinical trials within a CTU, had previous experience of qualitative interviewing and had a general interest in EDI in clinical research. Repeat interviews were not carried out, and transcripts were not returned to the participant for comment or correction. There were no other individuals present at the interviews and interviews lasted from 42 to 62 min. Participants undertook the interviewees either while at work or home. Field notes were not taken.

All interviews were undertaken via Google Meet, with the audio and video from the interview recorded (with consent) using in-built functionality within the Google Meet platform and transcribed for in-depth analysis. Transcripts were anonymised before analysis.

Emphasis was placed on collecting detailed data from experienced participants. Participants did not provide feedback on the findings. Patient and public interviewees were offered £50 as remuneration for their time.

## Relationships of interviewee and interviewers

For most interviewees, there was no prior relationship between the interviewees and the interviewers. However, interviewees would have been aware of the interviewer's interest in this research area. Two interviewees (DO and FT) were involved in the study as collaborators. They provided input into the study protocol and topic guides and were therefore more aware than the other participants of the background, rationale and aims of the study. However, they were not involved in the analysis and only provided feedback on the final written manuscript.

## Patient and public involvement

DO (a patient representative with experience of participating in research studies) provided input into all aspects of the study, including the aims and objectives, semi-structured topic guides and findings.

## Analysis

Data were analysed using thematic analysis, as described by Braun and Clarke.[36] NVivo software (V.12) was used to manage the data. RC and JW read two transcripts in order to familiarise themselves with their contents, then proceeding to independently generate codes through semantic analysis, deriving the themes from the data. Both researchers independently generated themes that related to the research questions and aims. RC and JW met to discuss the themes in order to ensure homogeneity. RC coded the remaining transcripts, adapting the codes where necessary. There were two separate coding trees for the two main participant types—professionals and patient and public representatives, with main themes including the barriers to incorporating diverse populations, the need for diversity in EPCTs, current practice and issues, thoughts on how to improve diversity, and any positive and negative effects of including diverse populations. The two coding trees were combined within the final analysis for readability as they included similar and complementing views on the same topics.

## RESULTS

### Study flow and characteristics of interviewees

68 individuals, based across 31 academic, healthcare and industry-focused institutions, were approached to participate in the study. Two individuals stated they did not want to take part due to a lack of time and five because of a lack of experience of EPCTs, as well as 53 who did not reply. Five professionals (based within the National Health Service (NHS) and at UK universities), and three patient and public representatives were interviewed— their characteristics are presented in table 1. All participants, except one patient and public representative, were based in the UK and had the experience of cancer EPCTs.

### Views on current practice

#### Lack of focus on recruiting underserved groups

Professional interviewees stated that they do not currently specifically focus on trying to recruit underserved groups to EPCTs.

I don't think it's something we consciously do. We basically open it to everyone who can enter into the

**Table 1** Participant characteristics

| Participant ID | Gender | Ethnicity* | Age range (years) | Role | Years of experience of early-phase trials† | Clinical area of experience |
|---|---|---|---|---|---|---|
| Participant 1 | Male | Not collected | 25-34 | Professional (operational lead at a CTU) | 6–10 | Cancer |
| Participant 2 | Female | Not collected | 45–54 | Professional (doctor) | >16 | Cancer |
| Participant 3 | Female | Black African | 55–64 | Patient and public representative | Not collected | Cancer |
| Participant 4 | Female | Not collected | 45–54 | Professional (statistician) | 11–15 | Cancer |
| Participant 5 | Male | Not collected | 35–44 | Professional (statistician) | 6–10 | Cancer |
| Participant 6 | Female | British Chinese | 65–74 | Patient and public representative | Not collected | Cancer |
| Participant 7 | Female | White Canadian | 35–44 | Patient and public representative | Not collected | Neurological |
| Participant 8 | Female | Not collected | 35–44 | Professional (nurse) | 11–15 | Cancer |

*Ethnicity only collected for the patient and public representatives.
†Only collected for professional participants.
CTU, clinical trials unit.

trial, but we do not actively go and look for underserved patients. (Participant 4, professional)

Those who did recall conversations on increasing diversity described rudimentary discussions around ensuring variation in the location of sites in the UK (eg, North or South), and providing translation services, rather than in-depth conversations around how to increase the diversity of their sample.

I think most of the time [discussions around diversity] will be about geography. It will be about "We've got a centre in London and I think I know someone up in Manchester. That'll be good because then we've got somewhere up north." (Participant 5, professional)

### Lack of diversity
It was evident that there is a lack of diversity in EPCTs, with three professionals describing the demographics of those who participate in EPCTs as being generally white, educated, affluent and young.

We know from our work here that patients coming through to our phase one trials are usually from the largely white, largely middle-class population. (Participant 2, professional)

### Awareness of need for diversity in EPCTs
Participants felt that awareness is increasing, with three interviewees discussing studies they are aware of looking at the level of diversity in EPCTs, and another participant recently becoming aware of a funder requesting information about diversity in grant applications.

The [EPCT] grant application I'm doing at the moment has three questions on [diversity], which we never had before. So, we're having to think about how we are going to address some of these issues. (Participant 1, professional)

### The level of importance for diversity in EPCTs
All interviewees stated that the inclusion of underserved groups is important and should always be considered when undertaking EPCTs.

You'd never want to be ignoring [diversity in EPCTs], I don't think. Trying to think on the spot why you would—I can't think of an argument why you'd not want to consider it. (Participant 1, professional)

Participants thought that there is a general need to increase diversity in EPCTs without specifically focusing on particular attributes, underserved groups or by setting a quota for underserved groups in the protocol.

I don't think you can start to tease out who you should and shouldn't exclude in a phase 1 trial, so we should include everyone. (Participant 2, professional)

I don't know if working on a quota system of, say, "We're recruiting 30 participants and we need, in this particular study, we need at least five to be black" or "We want a 50:50 split male and female". I don't know how practical that is in the early phase setting. (Participant 5, professional)

There were no situations when ensuring diversity may not be important. Even when undertaking an EPCT in a

specific genetic biomarker or disease (eg, prostate cancer in men), it would still be important to seek diversity in other characteristics (eg, socioeconomic status).

> [a] prostate cancer study obviously can only recruit males. If you wanted to look at an effect on a certain group or you did want to target a certain socioeconomic group then that could be an area where you'd target and exclude others. (Participant 1, professional)

### The need for diversity in EPCTs

Interviewees discussed diversity in EPCTs in terms of the need for exploring treatment signals in diverse populations, research equity, and informing robust design and conduct of later-phase trials.

#### Exploring treatment signals in diverse populations

The most frequently mentioned reason for increasing diversity in EPCTs was to ensure that EPCTs identify side-effects, tolerability and efficacy in diverse populations early, as different populations may react variably to the study treatment. Many interviewees stated the aim of including diverse populations is to get a 'feel' for the toxicities or a 'signal' of the efficacy in these populations.

> … in the early phase you're not going to get the statistics to demonstrate that this drug is what the additional toxicities are, or that a different dose should be given or anything like that. I think it's about getting a feel for what the broad categories of toxicities are and what the dose is. (Participant 2, professional)

#### Improve research equity

A key reason for diversifying clinical trial populations is to improve fairness and to enable underserved groups to access novel treatments.

> It's not fair that there are marginalised groups that are missing out on an opportunity to explore clinical trials and it might be that it's not the right thing for them and that might be the conclusion, but i'd much rather that patient have that informed decision about that before they make that decision. (Participant 8, professional)

Interviewees explained that incorporating diverse groups into EPCTs allows treatments to be identified that may only be efficacious in certain populations. Assuming that treatments that are not efficacious in a Caucasian population do not work for everyone is an injustice if the treatment in fact works in other populations.

> [If you undertake a] trial [of] a drug in a Caucasian population and it doesn't have any efficacy signals […] there's an injustice there for the underserved ethnic populations, where actually it could be an efficacious drug. (Participant 2, professional)

#### Informing robust design and conduct of later-phase trials

Improving the design and conduct of future trials was given as a reason to improve diversity, with specific considerations including understanding adherence to the new treatment in other populations, understanding how to better collect trial outcomes across diverse populations and identifying stratification variables for later-phase trials.

### Barriers to diversity and methods to improve diversity

Interviewees provided their views on the barriers to incorporating diversity, and associated methods to improve representation, across the following themes: the location of recruiting centres, recruitment practices, the extra resources needed to recruit underserved groups, participant perception (awareness and transparency) and a lack of information on diversity in EPCTs. A summary of the proposed barriers and associated resolutions is provided in table 2.

#### Location of recruiting centres in major cities

A main barrier to recruiting diverse populations was seen as the location of centres that undertake EPCTs, which are usually located in the centre of major cities. This can result in only those individuals who can travel to the centre being able to participate in the trial. This is compounded by the intensive time commitment required from participants to participate in these trials (eg, intensive follow-up schedules).

> I think the main challenge for early phase is actually the fact that it's a lot more intensive if you enter into a trial in the early phase setting, compared to later. If you look at the time schedule of how many times you have to go into clinics, there's a lot more of them in the early phase. (Participant 4, professional)

Two methods were suggested to improve access to the research site. First, both professionals and patient and public representatives suggested that participants and their carers or companions may need to be remunerated for their time (ie, lost earnings), subsistence and travel costs.

Second, referrals may be made from community hospitals. However, several participants stated that referrals from such centres often do not result in the recruitment of those from underserved groups due to participants often having to initiate the discussion regarding participation in an EPCT. To do so, participants require courage and conviction to ask their doctor about the trial, which may be less likely to be undertaken by those from underserved groups.

> Most of our patients that come to our unit are self-driven referrals [and] are well educated, articulate, confident, and would question their consultant and are driven by their own research into clinical trials and what's available and are driving their referral, … But what it does mean is that I feel we are not getting

**Table 2** Barriers and facilitators to increasing diversity in EPCTs

| Category | Barrier | Facilitator |
|---|---|---|
| Patient factors | EPCTs require intensive time commitment from trial participants, which may deter involvement | None discussed |
| | Participants may have to undertake 'self-driven' referrals and require the motivation and courage to initiate discussions regarding EPCTs with their clinician | ► Educate potential participants regarding the EPCTs that are available to them<br>► Educate clinicians to enable them to approach and recruit participants to EPCTs |
| | A perceived lack of transparency may result in those from underserved groups being unlikely to participate in an EPCT | ► Provide potential participants with testimonials from other participants<br>► Describe the need for diversity in the EPCT to those from underserved groups and the importance of participating |
| | It is important for participants to ask questions about the EPCT when discussing their potential participation, but participants may not ask questions, and clinicians may not encourage them, due to a lack of time | ► A list of potential questions can be given to participants to prompt them during a consultation with their doctor/clinician |
| | There is a lack of awareness of EPCTs among underserved groups | ► Discussions around participating in an EPCT should be initiated as early as possible, ideally by the GP<br>► Education to improve awareness, both for patients and clinical staff<br>► Permit use of trial databases to allow participants to locate EPCTs (eg, Experimental Cancer Medicine Centres in cancer)<br>► Patient advocacy groups could be established to assist in relaying information to underserved groups in a balanced manner |
| | Those from underserved groups may have concerns around the safety of participating in an EPCT | ► Communicate the level of risk associated with the EPCT with the participant<br>► Provide detailed information about what is already known about study treatment<br>► Provide detail as to what is unlikely to happen to participants (eg, death, paralysis)<br>► Educate patients to help improve their perception of EPCTs |
| Patient factors/ trial design and implementation | Many centres that recruit to EPCTs are located in the centre of major cities which may be challenging for certain populations to access | ► Referrals can be made from community hospitals<br>► Provide reimbursement to trial participants and their carers for travel and subsistence costs |
| Trial design and implementation | A lack of diversity in trial staff and researchers may deter participants from taking part | ► Increase the diversity of staff who undertake and run EPCTs |
| | Eligibility criteria may inadvertently reduce the diversity of the trial sample | ► Engage with a patient or diversity representative to review eligibility criteria |
| | Approaching underserved groups to participate in an EPCT requires extra resources | ► Additional funding and time may be required to recruit those from underserved groups |

EPCT, early-phase clinical trial; GP, general practitioner.

a representative population to even consider a clinical trial. (Participant 8, professional)

These actions were felt to improve access to the research site, but not the issue of the time commitment required of participants, an issue to which interviewees did not provide a resolution.

Even if you fund the travel, it's still difficult to have the time [to participate in an EPCT]. (Participant 4, professional)

**Unnecessary eligibility criteria**

The trial's eligibility criteria may indirectly reduce the diversity of the trial sample, which may be resolved by a patient or diversity representative reviewing the criteria to check for any elements that may inadvertently exclude individuals.

It might be that you're excluding people who have a certain risk of another particular condition, whether that's a problem with their heart or some other issue

… but what you're actually doing is you're making it far more likely that black women would not be eligible for that study because they're more likely to have that particular condition. Prior to the trial we engaged with an EDI representative to think about how we make this project as inclusive as possible. (Participant 5, professional)

### Participants need to be able to ask questions

Patient and public representatives felt it was important for the participant to be able to ask questions during the recruitment process. However, one professional participant stated many participants often do not feel able to do this, with clinicians also not encouraging questions due to a lack of time. One participant suggested that a sheet of prompts could be given to the participant to refer to during a consultation with their doctor to assist them in asking questions related to the EPCT.

I developed a question prompt list for patients to have in their hand and they could then refer to the questions that I developed with patients about what they felt they perhaps should have asked at the beginning. It gave the patient permission to ask those questions, [for example] emotive questions like "Will this trial cure my cancer? (Participant 8, professional)

### Extra resources are needed to recruit underserved groups

Approaching underserved groups to participate in EPCTs was seen as time-consuming. The use of translators to help those without English as their first language to comprehend the trial was seen as a key issue, with this adding to the cost and time demands.

Clinic appointments tend to take longer if there's an interpreter to make sure everything's been explained clearly and that the patient has a good understanding. And ultimately, if a clinic appointment takes longer and you have a high proportion of patients requiring interpreters for example, it potentially reduces the number of patients that you can recruit to a trial. (Participant 2, professional)

These issues may be intensified in deprived areas due to increased staff turnover and the reliance on locum doctors who lack the time or training to undertake recruitment processes. The increased workload required to recruit those from underserved populations does not result in increased funding.

Someone who's going to bring in patients from more diverse populations isn't going to generate any additional income, because we don't have a supply issue. And actually… [it] might decrease our income because patients who need a translator take longer in clinical time and might be able to treat fewer patients. (Participant 2, professional)

Therefore, interviewees recommended changes that funders could make, including providing trial teams with extended trial recruitment phases to allow time to recruit those from underserved populations. Additionally, extra per-participant funding for recruiting those from underserved groups may be required.

If you go down to the health economics of it, you've got to have a driver that will benefit us in some way and maybe that's some kind of funding from [the] NIHR [National Institute for Health and Care Research], where if a patient is labelled as being from a hard-to-reach population, that patient attracts additional income. (Participant 2, professional)

### Awareness and perception of EPCTs
#### Lack of awareness of EPCTs

Interviewees suggested that the awareness of EPCTs could be increased. This could be achieved by providing potential participants with access to databases that allow participants to be identified. For example, using a similar portal to the Experimental Cancer Medicine Centres trial finder for cancer EPCTs (which is currently only open to clinicians). Conversations about recruitment to EPCTs should be initiated as early as possible (ideally by the general practitioner). Patient advocacy groups could be used to assist in reaching out to underserved patient groups in a balanced manner.

I think it also requires quite a strong patient advocacy group of some sort that currently I don't think exists for EPCTs, which I think you need to reach out for those people to kind of reach out to people who don't normally think about entering into early phase trials. Kind of highlighting pros and cons, being very balanced. (Participant 4, professional)

#### Perception of EPCTs

Patient and public representatives focused on their perception of EPCTs. The safety of participating in an EPCT was seen as one of the main barriers to the inclusion of underserved groups, with the perceived risk being high, which was potentially exacerbated by the feeling of being a 'guinea pig'.

I think in the Chinese culture, and some others, they are normally equated as guinea pigs and you've got to be self-sacrificing, or prepared to be, before you think about something like this, something drastic might befall you. Which is rather scaremongering. (Participant 6, patient and public representative)

Several professional and patient and public representatives proposed that education may improve awareness and perception of EPCTs, with suggestions including the provision of general health education in schools, educating underserved groups and educating referrers in the need for diversity within EPCTs. However, there was a lack of suggestions on how to achieve this.

I think [education is] key. Educating referrers and somehow educating [underserved] groups about

 

the potential for accessing clinical trials. And I don't know fully how that's going to happen, how we can do that effectively. (Participant 8, professional)

Communicating the level of risk associated with participating in an EPCT was seen as key, with safety being a key concern for public representatives, who may want detailed information about what is already known about the drug.

### Improving transparency

Transparency was seen as being particularly important. Several methods of increasing transparency were suggested, including providing potential participants with testimonials from other participants and describing the need for diversity to those from underserved groups (ie, why they are seeking diverse populations to participate in the study and how they are including such groups).

One participant suggested that researchers should be honest if there is a lack of evidence regarding the safety of a new treatment (eg, a drug).

> If the researcher feels that "if I share with the potential participant how little actually I know, they will lose confidence." But I would argue that conversely, if you don't share with me what you know and what you don't know, and why am I involved, do I have any valid role in this? Can I actually make a contribution in some areas? So, in fact, transparency and that openness is confidence enforcing. (Participant 6, patient and public representative)

The need for transparency also extended to what *is not likely* to happen to a participant in an EPCT. One patient and public representative described that underserved populations have a lack of awareness of EPCTs

and may assume that there is a significant risk of death. Researchers should therefore clarify the likely adverse events the participant may experience, for example, clarifying that death or paralysis is very unlikely. This could take the form of describing how other people have reacted to the drug.

> Has somebody like me done it before, how did they react to it?… It's just reassuring me that it's gonna be okay, you know, you wouldn't be paralysed. (Participant 3, patient and public representative)

### Lack of research staff diversity

A lack of diversity was seen as not only a problem in the research participants themselves, but also in the staff and researchers who undertake and run such studies.

> The more diverse the people conducting the clinical trials, the more diverse it is, the better you see people forthcoming. "Oh, yes, I want to be part of that." When you don't see people like you in it, you know, what effect would it have on me? So, I know we're getting there but, you know, we still need to do more. (Participant 3, patient and public representative)

### Lack of information on diversity in EPCTs

Professional participants suggested that little is currently known about the current level of diversity within EPCTs, and therefore, postcode data could be collected for all participants screened for entry into an EPCT to identify if trials are being accessed by underserved groups. However, there may be ethical challenges in collecting such data for those participants who do not consent to participate in the trial.

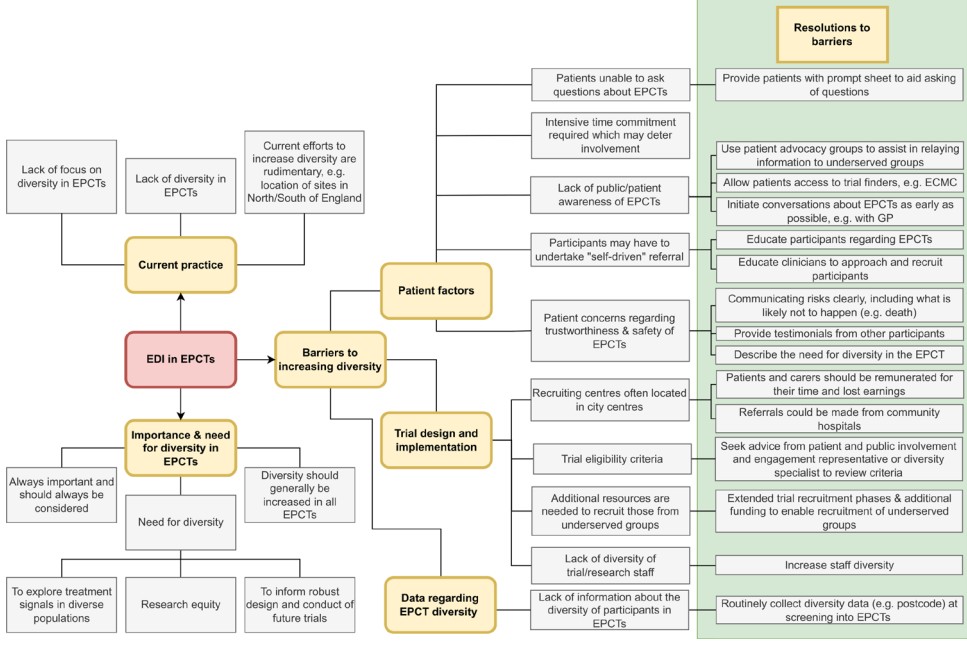

**Figure 1** Thematic schema. ECMC, Experimental Cancer Medicine Centres; EDI, equality, diversity and inclusion; EPCTs, early-phase clinical trials; GP, general practitioner.

## Box 1 Recommendations for researchers and funders

**Recommendations for researchers**

1. Increasing the diversity of the sample should always be considered in EPCTs to (a) explore treatment signals in diverse populations, (b) improve research equity, and (c) inform the robust design and conduct of later-phase trials. *Some diseases have specific diversity profiles, which should be reflected in the trial sample.*

2. Those undertaking EPCTs should make a concerted effort to recruit those from underserved groups. The diversity of EPCTs should generally be increased, rather than focusing on specific attributes or quotas.

3. Important markers of diversity (eg, postcode, *ethnicity, age*) should be consistently collected and reported from all those who are screened to participate in EPCTs.

4. In order to reach diverse communities, potential participants may be referred from community hospitals. However, participants may struggle to access the main trial site, so reimbursement for participants and their carers' time and travel costs should be considered. *Enhancing participant awareness of the reimbursement of travel and other costs is important.*

5. Education of both participants and staff is important to ensure both parties are aware of EPCTs, in order to enable recruitment and retention.

6. Researchers should ensure eligibility criteria do not inadvertently exclude those from underserved groups and should seek input from patient or diversity representatives into these criteria. *However, this should be balanced with the risk of the EPCT—some exclusions may be justified based on safety concerns.*

7. A list of prompts may be provided to participants to refer to during conversations regarding participation in an EPCT to assist them in asking questions about the study.

8. Discussions around entry into an EPCT should commence as early as possible in the participant's journey, ideally with the GP.

9. Participants from underserved groups may be concerned about the risks of participating in an EPCT. In order to reassure them, researchers may:
   a. Clearly communicate the level of risk associated with the EPCT with the participant. The societal benefits of participating in an EPCT may also be relayed to participants (see principle 1).
   b. Provide detailed information about what is already known about the study treatment.
   c. Provide an honest explanation of the amount available regarding the evidence regarding the effects of the study treatment.
   d. Provide information as to what is *unlikely* to happen to participants as a result of the study treatment (eg, death, paralysis).

10. In order to assist in relaying information to underserved groups in a balanced manner, patient advocacy groups may be formed.

11. Due to a perceived lack of transparency, those from underserved groups may be provided with testimonials from other participants. The need for diversity in the EPCT may be explained to such individuals.

12. The diversity of those staff who recruit to EPCTs should be increased in order to not deter underserved groups from participating.

**Recommendations for funders**

13. Funders may need to provide additional per-participant funds for the recruitment of those from underserved groups due to the additional resources required to recruit such individuals.

14. Funders should request details of the methods investigators will use to identify, approach and recruit those from underserved groups in grant applications.

EPCT, early-phase clinical trial; GP, general practitioner.

A schema, depicting the themes and the associations between them, is provided in figure 1.

## DISCUSSION

In this qualitative study, we have identified that the diversity of underserved groups does not seem to be currently routinely considered in EPCTs, but it is always viewed as relevant. The main benefit of including diverse populations in EPCTs is the identification of early signals of harms and efficacy in diverse populations, including underserved subpopulations, and providing equitable opportunities for patients. Study teams should make a concerted effort to recruit those from underserved groups into their EPCT and should seek to minimise the numerous barriers that prevent such individuals from participating. The most notable barrier appeared to be that EPCTs are primarily undertaken at a small number of specialist centres, predominately located within the centre of large cities, thus limiting the populations they can access. To increase diversity, these centres may need to rely on referrals from 'feeder' centres located in the community. However, the extra resources it requires to approach and recruit underserved populations (eg, language translation) can prevent clinicians from referring them. Patient and public representatives stated that trust and communication are important—those recruiting underserved groups to EPCTs should be transparent regarding the need for diversity in clinical trials, the potential harms of the treatment and if these are not known, clarifying the likelihood of severe illness to ease their concerns. Education of both patients/public and clinical staff was felt to be key to increasing awareness and perception of EPCTs.

The strengths of this study are that both professionals and patient and public representatives were sought to obtain the views of two of the key stakeholders involved in EPCTs. Both sets of individuals were diverse, in terms of the role of professional participants (including clinicians, statisticians and operational leads) and the ethnicity of the patient and public representatives. Due to funding limitations, only a small number of interviews were undertaken. Additionally, the approvals necessary to interview staff in the NHS were not sought due to time limitations, and therefore only those staff who also had contracts with a university were interviewed, thus limiting the sample to those NHS staff who also had academic posts. Those from other organisations (eg, the pharmaceutical industry) were not interviewed. Additionally, all except one interviewee's experience was primarily in UK-based cancer EPCTs, therefore limiting the conclusions that can be made regarding other disease areas and countries. Funding was not available to undertake additional recruitment to interview those with expertise outside of cancer trials research. There was no provision for those individuals who required assistance to converse in English,

which prohibited those with limited English proficiency from participating in this study. Lastly, ethnicity data were collected for public and patient only, and therefore we are unable to report and reflect on the ethnicity of 'professional' participants.

This study supports the literature regarding the need to increase diversity in clinical trials.[6 37] Some of the barriers and methods to increase diversity have been reported previously across both early and late-phase trials, including adequately funding participants' time and travel costs,[38] restrictive eligibility criteria,[12] burdensome trial procedures,[12] informing community oncologists of the trial to aid recruitment,[32] and translation and language issues.[30] We have identified that, due to time constraints, those who refer to clinical trials may not approach those from underserved groups, which has been found in a previous study.[39] Our study adds to this literature by clarifying that diversity is important in all EPCTs, and by identifying additional barriers that, as far as we are aware, have not been previously reported. These additional barriers concern the effect of the location of EPCT units on their ability to access underserved groups and the lack of referrals to these units of underserved groups from referrers in the community. Such referrals may be hindered by the need for participants to 'self-drive' the referral, which may be a barrier given the lack of knowledge of EPCTs in underserved groups. We have also identified specific issues related to transparency, including informing the participant the reason for the need to focus on the recruitment of underserved groups. Interviewees also suggested that an indication of the level of risk associated with the EPCT is provided during the recruitment process, even if the effects of the drug are unknown. It is important to note that, as per Good Clinical Practice guidelines, the risks of participating in research should always be communicated to patients in a culturally responsive way.[40] However, participants in this study desired an estimation of the risks of research even when these cannot be quantified, for example, reassurance that the drug under test is unlikely to lead to death or significant injury.

This study implies that those designing and recruiting to EPCTs should attempt to increase the diversity of their sample by referring to the 14 principles outlined in Box 1. In this box, the authors have supplemented the findings from the qualitative study with their own reflections, which are provided in *italics.*

It should be noted that given that those participating in EPCTs may have optimistic expectations of the effect of participating in an EPCT,[27] highlighting the harms of such trials may have a negative effect on recruitment, so researchers may also describe to participants the potential societal benefits of EPCTs highlighted in this study (eg, improving the equity of research, exploring early treatment signals in diverse populations and informing the robust design of future trials). Future research should focus on exploring the barriers and enablers to recruitment in a larger diverse sample of participants, including NHS staff, and by investigating how potential harms are

communicated to participants, especially in the absence of quantifiable data. Overcoming the inherent barriers to undertaking clinical trials in novel treatments, which, due to their risk, often require participants having to regularly attend the hospital for safety checks, may be challenging but would open up EPCTs to patient groups who are currently unable to participate in such trials. These 'remote' EPCTs have been broached, but are rarely undertaken in practice.[41] Additionally, educational packages could be developed to increase awareness and improve the perception of EPCTs in underserved groups—future research may focus on the content, audience, timing and format of such packages.

**Author affiliations**
[1]Sheffield Clinical Trials Research Unit (CTRU), Sheffield Centre for Health and Related Research, The University of Sheffield, Sheffield, UK
[2]Patient representative, London, UK
[3]The Christie Hospital NHS Foundation Trust and University of Manchester, Manchester, UK
[4]Clinical Trials and Statistics Unit, The Institute of Cancer Research, Sutton, UK
[5]Department of Infection, Immunity, and Cardiovascular Disease, The University of Sheffield and Sheffield Teaching Hospitals NHS Foundation Trust, Sheffield, UK

**Acknowledgements** We would like to thank the eight participants for their time in participating in the interviews and for sharing their valuable knowledge with us. We remember Dolapo (Della) Ogunleye, an integral member of this study team, who passed away before this manuscript was published. Her passion, insights and dedication to improving diversity and inclusion of participants in research to achieve equitable patient care enriched every aspect of this work.

**Contributors** RC, MD and CC devised the study. RC, SD and JW collected the data and undertook the analysis. RC wrote the first draft of the report. DO was a patient representative who provided input into the design of the study and the findings. All authors, including AR, CY, FT and KB, provided input into the design and implementation of the study, critically reviewed the draft and provided approval for publication. RC is the guarantor of the work.

**Funding** The study was funded by Research England via the Participatory Research Fund (X/175529).

**Competing interests** None declared.

**Patient and public involvement** Patients and/or the public were involved in the design, or conduct, or reporting, or dissemination plans of this research. Refer to the Methods section for further details.

**Patient consent for publication** Not applicable.

**Ethics approval** This study involves human participants and was approved by School of Health and Related Research (ScHARR), within the University of Sheffield (reference number: 045712). Participants gave informed consent to participate in the study before taking part.

**Provenance and peer review** Not commissioned; externally peer reviewed.

**Data availability statement** No data are available. The dataset is the in-depth interview transcripts. Access is restricted to the study team in accordance with the requirements of the ethical approval.

**ORCID iDs**
Robin Chatters http://orcid.org/0000-0002-1945-6011
Katie Biggs http://orcid.org/0000-0003-4468-7417
Christina Yap http://orcid.org/0000-0002-6715-2514

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
