## [Reviewer comments · BMJ Open]

ARTICLE DETAILS

TITLE (PROVISIONAL)	Exploring the barriers to, and importance of, participant diversity in early phase clinical trials: an interview-based qualitative study of professionals and patient and public representatives
AUTHORS	Chatters, Robin; Dimairo, Munyaradzi; Cooper, Cindy; Ditta, Shamila; Woodward, Jonathan; Biggs, Katie; Ogunleye, Della; Thistlethwaite, Fiona; Yap, Christina; Rothman, Alexander

VERSION 1 – REVIEW

REVIEWER	Ivarsson, Melanie Moderna Inc
REVIEW RETURNED	06-Jul-2023

GENERAL COMMENTS	This is an interesting paper highlighting an important topic. I would appreciate it if the Authors could consider the following points: 1. The sample size of interviewees is very small. While the Authors justify this as part of their methods, they had originally reached out to many more potential participants. The interviewees are also from a similar academic background (in addition to the participant interviewees). There was also a limited geographic spread of interviewees (from the same institution?). This leads the Reader to speculate that the research itself was done in a way that was not diverse or inclusive. Can the Authors discuss this further, highlighting how they would have conducted this research if funding had allowed or more people had responded to the invitation? What efforts could have been made to include more diverse participation and how could they have included a wider range of backgrounds and perspectives in this qualitative survey?2. In the Introduction the Authors describe the importance of conducting EPCTs in diverse populations, however they do not describe any evidence of why this has shown to be important in the case of interventional clinical trials and provide only speculative evidence for this. Can the Authors confirm whether they have conducted a review of the literature on this topic and their findings. This may help boost the argument on the importance of diversity in EPCTs.3. All EPCTs should be conducted with the appropriate Informed Consent process in place according to GCP. The recommendations within this paper do not refer to this CT process but mention the need for patients to be informed of the risks. Please include a reference to the importance of appropriate consent processes.
---

	Overall, the paper is an interesting discussion of the barriers to inclusion in ECPTs. However the rationale for why this is important needs to be strengthened. Ideally a more diverse group of participants should have been sampled for this study. The conclusions in this paper are widely known and understood today as the barriers to inclusion in Clinical Trials overall. The Authors may want to consider strengthening the argument for why this is important in EPCTs.
--	---

REVIEWER	Cervantes, Lilia University of Colorado - Anschutz Medical Campus
REVIEW RETURNED	12-Oct-2023

GENERAL COMMENTS	Thank you for the opportunity to review “A qualitative interview study exploring the barriers and importance of participant diversity in early phase clinical trials” for BMJ Open. Overall this manuscript offers a unique perspective on the need for diversity in clinical trials and the authors are commended for their efforts. Additional consideration as to the study design and analysis as discussed below. Recommendations: Major:  • The small sample size hinders the study’s findings, and in addition, the viewpoints seem to be larger limited to cancer trials. Further reasoning needs to be given why additional recruitment was not considered. While the argument for maximal variation is considered, the viewpoints do not reflect clinical trials in other disease states. This should also be listed as a limitation. • Please define how two separate coding trees for two different groups of interviewed participants became a combined thematic analysis, or why it was felt the results could be combined. - Methods:  o Regarding the interview guide, please describe if there was a literature review (and provide citations) when developing the interview guide and if it was iteratively modified. Provide a supplement with the questions. o It is unclear how purposive sampling was pursued for this study since there were only 8 individuals that responded to the survey and agreed to participate. There weren’t any additional individuals who filled out the survey who might have been eligible and who were selected/not selected. Please clarify. o Regarding research ethics, were people compensated financially for their time to participate and was there consideration for inclusion of individuals who report limited English proficiency. o Regarding sample size, why was data saturation not considered? I think its important to be transparent and perhaps mention that data saturation was not possible because of the limited number of participants. As a reviewer, I want to understand if there were additional efforts undertaken to recruit more participants to achieve data saturation. o The fact that two interviewees were involved in the study as collaborators is unusual and is concerning with respect to bias -- 2 of 8 (1/4th) of participants were essentially involved in the analysis/drafting of the manuscript which is concerning with respect to bias of results. This means that there was only 2 participants who represented the patient/public for this study. o Please include the initials of the two interviewees that were also study collaborators—if they were involved in thematic analysis, this is problematic.
--

	o Another concern is the use of 3 interviewers for 8 individuals. With each likely interviewing 2 to 3 people, there's little experience to iteratively modify the questions and dive deeper into the experiences individuals are sharing.  • Please further define the methodology used throughout the coding process • The thematic analysis needs to be reviewed and refined further. Specific recommendations:  o Were there sub-themes for the first two themes? Reading through it appears there were, for example, "Views of current practice" could be broken into "lack of or rudimentary recruitment", "lack of diversity in prior trials" and "awareness of need for diversity in EPCTs". o There are too many sub themes under "barriers to diversity and methods to improve diversity". Consider refining coding or reviewing the thematic analysis. o "Awareness and perception of EPCTs" is incredibly long for a sub-theme and appears to have multiple sub-themes, so it may be considered making this its own theme entirely. o Consider reviewing the data and renaming the themes to more accurately describe (not categorize) the results. For example, "views on current practice" does not provide any insight as to what the participants said. Another example is "recruitment practices and trial design" sub-theme – this does not describe the issues addressed in the body of the results. • Ensure that the results are limited to what participants stated. For example, there is a whole paragraph explaining why referrals from community hospitals would be difficult (starting with "However, referrals from such centres...") If these were specifically called out by participants, this should be mentioned, otherwise this should be saved for discussion. This is noted several times throughout the results. • Would recommend a table with additional illustrative quotes listed by theme and sub-theme. This may streamline the results section. • Consider a thematic schema to tie the themes together. • Limitations need to be addressed in the discussion. • Table 2: Consider utilizing a framework to organize table 2 into levels of barriers and facilitators (ie, institutional level OR stage of EPCT recruitment, dissemination etc), as it is hard to read without clear organization Minor:  • Results in the abstract should be listed by theme. • Evaluate the use of equality instead of equity. • There are different size fonts for different sections. For example, "Barriers to Diversity and Methods to improve diversity" is smaller than "The need for diversity in EPCTs". • Table 1: why was race/ethnicity not collected uniformly? • Please include the interview guide as a supplemental table
--	---

VERSION 1 – AUTHOR RESPONSE

REVIEWER 1:

COMMENT: The sample size of interviewees is very small. While the Authors justify this as part of their methods, they had originally reached out to many more potential participants. The interviewees are also from a similar academic background (in addition to the participant interviewees). There was also a limited geographic spread of interviewees (from the same institution?). This leads the Reader

to speculate that the research itself was done in a way that was not diverse or inclusive. Can the Authors discuss this further, highlighting how they would have conducted this research if funding had allowed or more people had responded to the invitation? What efforts could have been made to include more diverse participation and how could they have included a wider range of backgrounds and perspectives in this qualitative survey?

RESPONSE: We thank the reviewer for their comments and agree that the sample size is small. We looked to recruit individuals across the spectrum of early phase trials, regardless of geographical location. We approached a total of 68 individuals from 31 institutions. Unfortunately, due to funding restraints, we did not have the time to seek the necessary approvals to interview NHS staff, and therefore, were limited to interviewing those staff that also had substantive University contracts. The reviewer is correct that two of the individuals were from the same institution, but there was a good spread of interviewees from other institutions. As suggested, we have added a section to the discussion section to explore the issue of diversity in this study, see page 17:

“The approvals necessary to interview staff in the National Health Service (NHS) were not sought due to time limitations, and therefore only those staff that also had contracts with a University were interviewed, thus limiting the sample to those NHS staff that also had academic posts.”

And we have also added, also on page 17:

“Lastly, there was no provision for those individuals who required assistance to converse in English, which may have prohibited those with significant barriers to taking part in EPCTs from participating in this study.”

We have also added further description as to the sample of individuals we approached to the results section (page 7) in order to highlight the diversity of individuals that were approached to participate in this study:

“Sixty-eight individuals, based across 31 academic, health-care and industry focussed institutions, were approached to participate in the study.”

COMMENT: In the Introduction the Authors describe the importance of conducting EPCTs in diverse populations, however they do not describe any evidence of why this has shown to be important in the case of interventional clinical trials and provide only speculative evidence for this. Can the Authors confirm whether they have conducted a review of the literature on this topic and their findings. This may help boost the argument on the importance of diversity in EPCTs.

RESPONSE: We have conducted a literature review on this topic. A few important manuscripts have been published since we initially drafted this paper, which we have included in the second paragraph of the introduction. We have found that the evidence in this area is lacking specific guidance for early phase clinical trials (EPCTs), with much of the evidence being non-specific to clinical trials in general.

COMMENT: All EPCTs should be conducted with the appropriate Informed Consent process in place according to GCP. The recommendations within this paper do not refer to this CT process but mention the need for patients to be informed of the risks. Please include a reference to the importance of appropriate consent processes.

RESPONSE: Thank you, this is an important point. We have added this to the discussion section, page 18: “It is important to note that, as per Good Clinical Practice (GCP) guidelines, the risks of participating in research should always be communicated to patients.”

COMMENT: Overall, the paper is an interesting discussion of the barriers to inclusion in ECPTs. However the rationale for why this is important needs to be strengthened. Ideally a more diverse group of participants should have been sampled for this study. The conclusions in this paper are widely known and understood today as the barriers to inclusion in Clinical Trials overall. The Authors may want to consider strengthening the argument for why this is important in EPCTs.

RESPONSE: It's important to note that the identified barriers are common across all clinical trial phases - we have added clarification regarding this to the discussion section, page 17: "Some of the barriers and methods to increase diversity have been reported previously across both early and late phase trials"

We have also strengthened the argument as to why it's important to explore the barriers to diversity in EPCTs to the background section by adding detail regarding the lack of literature in this area on page 3: "Two reviews and a Food and Drug Administration (FDA) white paper provide guidance around increasing clinical trial diversity in the drug development process, but all lack guidance specific to EPCTs"

REVIEWER 2

COMMENT: The small sample size hinders the study's findings, and in addition, the viewpoints seem to be larger limited to cancer trials. Further reasoning needs to be given why additional recruitment was not considered. While the argument for maximal variation is considered, the viewpoints do not reflect clinical trials in other disease states. This should also be listed as a limitation.

RESPONSE: We agree that the findings of the study are largely limited to cancer trials. We tried to recruit additional interviewees with experience of non-cancer trials but such individuals did not agree to participate. Unfortunately time and funding were limited so we were unable to extend the timelines of the study in order to increase variation and recruit additional participants. We have expanded upon this limitation within the discussion section on page 17: "Unfortunately, funding was not available to undertake additional recruitment to interview those with expertise outside of cancer trials research."

COMMENT: Please define how two separate coding trees for two different groups of interviewed participants became a combined thematic analysis, or why it was felt the results could be combined.

RESPONSE: The themes (or coding trees) from the two participant groups (patient representatives and professionals) were combined as they had similar, complementing, or sometimes different views on the same topics. It therefore made sense to cover the views of different participant groups within each theme at the same time within the manuscript, rather than describe their views in two distinct sections, which would have read quite repetitively. The reason for combining the two coding trees within the manuscript has been added to the methods section, see page 7: "The two coding trees were combined within the final analysis for readability as they included similar and complementing views on the same topics."

COMMENT: Methods:

o Regarding the interview guide, please describe if there was a literature review (and provide citations) when developing the interview guide and if it was iteratively modified. Provide a supplement with the questions.

RESPONSE: This detail has been added to the methods section of the manuscript (page 6): "A literature review was undertaken to inform the topic guide, identifying a lack of guidance specifically relating to EPCTs. As a result, literature regarding EDI in later phase trials (III or IV) was used to inform the interview questions."

COMMENT: It is unclear how purposive sampling was pursued for this study since there were only 8 individuals that responded to the survey and agreed to participate. There weren't any additional individuals who filled out the survey who might have been eligible and who were selected/not selected. Please clarify.

RESPONSE: Twenty-two individuals completed the survey, from which the interviewees were purposefully sampled. This detail has been added to the methods section of the manuscript (page 5): "Participants were selected purposefully from those that completed the survey (n=22)."

COMMENT: Regarding research ethics, were people compensated financially for their time to participate and was there consideration for inclusion of individuals who report limited English proficiency.

RESPONSE: Patient and public contributors were provided with £50 for taking part in the interview. Professional participants were not remunerated for their time. This detail has been added to the methods section, see page 6: "Patient and public interviewees were offered £50 as remuneration for their time." Unfortunately there was no provision for those that reported limited English proficiency, which we have added to our limitations section (see page 17):

"Lastly, there was no provision for those individuals who required assistance to converse in English, which may have prohibited those with significant barriers to taking part in EPCTs from participating in this study."

COMMENT: Regarding sample size, why was data saturation not considered? I think its important to be transparent and perhaps mention that data saturation was not possible because of the limited number of participants. As a reviewer, I want to understand if there were additional efforts undertaken to recruit more participants to achieve data saturation.

RESPONSE: Data saturation was never considered due to the small amount of funding available, limited time, and the perceived high level of variation in interviewee's experiences and viewpoints. In order to seek saturation a much larger study is required. We agree with the reviewer that this detail should be added to the manuscript, which we have added to page 5: "Due to the potential large sample size required to achieve data saturation, and the funding and time limitations of the study, data saturation was not considered"

COMMENT: The fact that two interviewees were involved in the study as collaborators is unusual and is concerning with respect to bias -- 2 of 8 (1/4th) of participants were essentially involved in the analysis/drafting of the manuscript which is concerning with respect to bias of results. This means that there was only 2 participants who represented the patient/public for this study.

RESPONSE: We agree with the reviewer that including two collaborators as interviewees is unusual. However, we disagree that this is concerning and has led to bias. The collaborators were not directly involved in the analysis, i.e., they were not involved in any of the analysis steps as described by Braun and Clarke (e.g., familiarisation, coding etc). They did not have access to the transcripts and their only involvement was checking the findings described in the manuscript met their interpretation. Therefore, this can be described as "member checking" for a limited number of interviewees. Detail regarding this has been added to page 6: "However, they were not involved in the analysis and only provided feedback on the final written manuscript."

COMMENT: Please include the initials of the two interviewees that were also study collaborators—if they were involved in thematic analysis, this is problematic.

RESPONSE: We have included the initials of the study collaborators in the results section. As described above they were not involved in the analysis.

COMMENT: Another concern is the use of 3 interviewers for 8 individuals. With each likely interviewing 2 to 3 people, there's little experience to iteratively modify the questions and dive deeper into the experiences individuals are sharing.

RESPONSE: The primary author (RC) led seven of the interviews. All interviews were also attended by another researcher (either SD or JW) to assist in ensuring all relevant questions on the topic guide had been asked. This detail has been added to the methods section of the manuscript (page 6): "Seven of the eight interviews were conducted by RC (a male Research Fellow with a BSc), with assistance from JW (a male Research Assistant with an MSc) and SD (a female Research Assistant with an MA). One interview was conducted by SD and JW."

COMMENT: Please further define the methodology used throughout the coding process

RESPONSE: Additional information has been added regarding the coding process (page 7): “RC and JW read two transcripts in order to familiarise themselves with their contents, then proceeding to independently generate codes through semantic analysis, deriving the themes from the data. Both researchers independently generated themes that related to the research questions and aims. RC and JW met to discuss the themes in order to ensure homogeneity. RC coded the remaining transcripts, adapting the codes where necessary.”

COMMENT: The thematic analysis needs to be reviewed and refined further. Specific recommendations:

o Were there sub-themes for the first two themes? Reading through it appears there were, for example, “Views of current practice” could be broken into “lack of or rudimentary recruitment”, “lack of diversity in prior trials” and “awareness of need for diversity in EPCTs”.

RESPONSE: Yes the reviewer is correct that this section consists of several sub-themes. Headings have been added accordingly.

COMMENT: There are too many sub themes under “barriers to diversity and methods to improve diversity”. Consider refining coding or reviewing the thematic analysis.

RESPONSE: This section has been split into its constituent sub-themes.

COMMENT: “Awareness and perception of EPCTs” is incredibly long for a sub-theme and appears to have multiple sub-themes, so it may be considered making this its own theme entirely.

RESPONSE: We agree that this theme is too long. We have separated ‘awareness and perception of EPCTs’ into its constituent sub-themes.

COMMENT: Consider reviewing the data and renaming the themes to more accurately describe (not categorize) the results. For example, “views on current practice” does not provide any insight as to what the participants said. Another example is “recruitment practices and trial design” sub-theme – this does not describe the issues addressed in the body of the results.

RESPONSE: We agree with the reviewer’s comment and have altered the headings accordingly.

COMMENT: Ensure that the results are limited to what participants stated. For example, there is a whole paragraph explaining why referrals from community hospitals would be difficult (starting with “However, referrals from such centres...”) If these were specifically called out by participants, this should be mentioned, otherwise this should be saved for discussion. This is noted several times throughout the results.

RESPONSE: All results are based on the participants' views. The results section has been altered to reflect this.

COMMENT: Would recommend a table with additional illustrative quotes listed by theme and sub-theme. This may streamline the results section.

RESPONSE: We do not believe this would improve the results section and benefit the manuscript. The results section already contains illustrative quotations. The manuscript is already fairly long and contains two large tables, as well as the schema as suggested by the reviewer.

COMMENT: Consider a thematic schema to tie the themes together.

RESPONSE: This has been added to the manuscript, please see Figure 1.

COMMENT: Limitations need to be addressed in the discussion.

RESPONSE: Limitations were addressed in the discussion, please see the second paragraph. However, we have added additional limitations to this section following peer-review.

COMMENT: Table 2: Consider utilizing a framework to organize table 2 into levels of barriers and facilitators (ie, institutional level OR stage of EPCT recruitment, dissemination etc), as it is hard to read without clear organization

RESPONSE: We have made the suggested changes to table 2.

COMMENT: Results in the abstract should be listed by theme.

RESPONSE: The abstract currently describes all the main themes identified in the study. We are unsure what the reviewer means by "listing by theme" - is this about using the name of the theme in the abstract? If so, we have tried to write the abstract in this manner but this affected readability.

COMMENT: Evaluate the use of equality instead of equity.

RESPONSE: We are unsure if the reviewer is suggesting that either 'equality' or 'equity' should be used. We believe that equality is the correct terminology, as this has been used across many reports and manuscripts within the acronym "EDI". There are a couple of uses of the word "equity" in the results section. This term has been used in this section as it was used by participants in the interviews.

COMMENT: There are different size fonts for different sections. For example, "Barriers to Diversity and Methods to improve diversity" is smaller than "The need for diversity in EPCTs".

RESPONSE: This has been corrected.

COMMENT: Table 1: why was race/ethnicity not collected uniformly?

RESPONSE: It was felt unnecessary to collect the race/ethnicity of professional participants, as there was not a focus on recruiting professional participants from ethnic minority backgrounds. In line with data protection principles, collecting such data from these individuals was seen as unnecessarily and potentially intrusive. However, the recruitment of public and patient representatives from ethnic minority backgrounds was a focus, so this data was collected from those individuals.

COMMENT: Please include the interview guide as a supplemental table

RESPONSE: This has been included.

VERSION 2 – REVIEW

REVIEWER	Cervantes, Lilia University of Colorado - Anschutz Medical Campus
REVIEW RETURNED	12-Jan-2024

GENERAL COMMENTS	Thank you for responding to all reviewer comments. Major: - In abstract, please modify 2nd bullet to "Data saturation was not achieved due to a small sample size" Minor: - In the second paragraph re evidence of importance of conducting EPCTs in diverse populations, could include some of the data from COVID-19 EPCTs and how this influence uptake/mistrust by diverse populations. Also consider including underrepresentation of individuals with limited English proficiency. - Minor change to added sentence on page age. Add "culturally responsive" to "It is important to note that, as per Good Clinical Practice (GCP) guidelines, the risks of participating in research should always be communicated to patients in a CULTURALLY RESPONSIVE way."
---

	 - Please replace 'unfortunately' throughout the manuscript with a different word - On page 17, consider modifying the sentence describing the lack of participation from individuals who report limited English proficiency. Consider "...which prohibited those with limited English proficiency to participate in this study." - Regarding data saturation, it is possible to achieve this with a small sample size and this depends on the topic and target population. It is premature to say that a large sample size was required. Please modify page 5 and remove "due to the potential large sample size required" and replace with "Data saturation was not achieved due to limitations in funding." - I disagree that including two participants as collaborators is 'member-checking' but clarifying that they were not involved in the analysis, as you have added, is important. - please include the lack of collection of race and ethnicity of participants in limitations section.
--	---

VERSION 2 – AUTHOR RESPONSE

Reviewer: 2:

Comment: In abstract, please modify 2nd bullet to "Data saturation was not achieved due to a small sample size"

Response: This change has been made as suggested, please see page 3.

Comment: In the second paragraph re evidence of importance of conducting EPCTs in diverse populations, could include some of the data from COVID-19 EPCTs and how this influence uptake/mistrust by diverse populations. Also consider including underrepresentation of individuals with limited English proficiency.

Response: Thank you for this suggestion. We have added this information to the introductory section, please see page 4.

Firstly, we have added limited proficiency in English to the list of factors that EPCTs underrepresent: "EPCTs underrepresent women [13], older adults [14–16], those with lower education levels [17], those who live further away from hospitals [17], those from deprived areas [18], ethnic minorities [19,20] and, for those studies undertaken in majority English speaking societies, individuals with limited English proficiency."

Secondly, we have added a sentence that highlights the evidence obtained from COVID-19 EPCTs: "The development and deployment of COVID-19 vaccines highlighted the role of medical mistrust in preventing individuals from participating in EPCTs, especially in those from ethnically diverse communities."

Comment: Minor change to added sentence on page age. Add "culturally responsive' to "It is important to note that, as per Good Clinical Practice (GCP) guidelines, the risks of participating in research should always be communicated to patients in a CULTURALLY RESPONSIVE way."

Response: Thank you. This change has been made.

Comment: Please replace 'unfortunately' throughout the manuscript with a different word

Response: We have removed the word 'unfortunately' throughout the manuscript. This word was superfluous so we have not replaced it with a different word.

Comment: On page 17, consider modifying the sentence describing the lack of participation from individuals who report limited English proficiency. Consider "...which prohibited those with limited English proficiency to participate in this study."

Response: This change has been made, as suggested.

Comment: Regarding data saturation, it is possible to achieve this with a small sample size and this depends on the topic and target population. It is premature to say that a large sample size was required. Please modify page 5 and remove "due to the potential large sample size required" and replace with "Data saturation was not achieved due to limitations in funding."

Response: This change has been made, as suggested.

Comment: please include the lack of collection of race and ethnicity of participants in limitations section.

Response: Thank you for your comment. We have added the following text to the limitations section on page 17: "Ethnicity data was collected for public and patient only, and therefore we are unable to report and reflect on the ethnicity of 'professional' participants."

VERSION 3 – REVIEW

REVIEWER	Cervantes, Lilia University of Colorado - Anschutz Medical Campus
REVIEW RETURNED	20-Feb-2024

GENERAL COMMENTS	The authors have adequately responded to revisions.
---